# Antimicrobial effect of oral care gel containing hinokitiol and 4-isopropyl-3-methylphenol against intraoral pathogenic microorganisms

**Hiroshi Ohara**[1,2]*, **Keita Odanaka**[1], **Miku Shiine**[1], **Masataka Hayasaka**[1]

**1** Department of Clinical Pharmacy, School of Pharmaceutical Sciences, Ohu University, Koriyama, Japan,
**2** Department of Pharmacy, Ohu University Hospital, Koriyama, Japan

* h-ohara@pha.ohu-u.ac.jp

**Data Availability Statement:** All relevant data are within the paper.

**Funding:** Initials of the authors who received each award: Not applicable Grant numbers awarded to

## Abstract

### Objective

Deterioration of oral hygiene is closely related to an increase in severity and mortality of corona virus disease-19 (COVID-19), and also contributes to the development of various diseases such as aspiration pneumonia or Alzheimer's. Oral care is attracting high interest in Japan, which has entered a super-aging society. In this study, we aimed to investigate whether commercially available Hinora® (HO), an oral care gel containing hinokitiol and 4-isopropyl-3-methylphenol (IPMP), has biofilm formation inhibitory and antimicrobial activities against various intraoral pathogen microorganisms.

### Method

*Candida* species, *Aggregatibacter actinomycetemcomitans*, *Staphylococcus aureus*, and *Pseudomonas aeruginosa* were selected during the study period, all of which were analyzed using antimicrobial disc, microorganism turbidity, and crystal violet assays. In addition, the germ tube test using *Candida albicans* (*C. albicans*) was performed with a modification of Mackenzie's method. Images for morphological observation of the germ tubes were acquired using an inverted microscope. For comparison between products, we used Refrecare® (RC), which only contains hinokitiol (not containing IPMP).

### Results

All the intraoral pathogenic microorganisms showed drug susceptibility against undiluted forms of HO and/or RC. In particular, HO was more effective at lower concentrations than RC. In the HO-added group, inhibition circles were observed in all bacteria except *P. aeruginosa* when added at a concentration of 0.5 g/mL or more. The optical density values at 590 nm (crystal violet) and/or 600 nm (microorganism turbidity) of all the fungi and bacteria were significantly lower when cultured in medium with HO. Inhibition of growth or biofilm formation was observed when HO was added at a concentration of 0.05 g/mL or higher. To investigate the action mechanism of HO, germ tube tests were performed in *C. albicans*. The results

each author: There is no grant number assigned to each author. The full name of each funder: Kenichi Ogasawara and Ryosuke Kawawaki URL of each funder website: https://www.enotsuka.co.jp/ Did the sponsors or funders play any role in the study design, data collection and analysis, decision to publish, or preparation of the manuscript? The funders had no role in study design, data collection and analysis, decision to publish, or preparation of the manuscript.

**Competing interests:** I have read the journal's policy and the authors of this manuscript have the following competing interests: We received a research grant from Otsuka Pharmaceutical Co., Ltd. to conduct and advance this research.

showed that culturing *C. albicans* in soybean-casein digest broth with HO (0.05 g/mL) significantly suppressed germ tube formation.

## Conclusions

These data suggest that oral care gel-containing hinokitiol and IPMP has strong biofilm formation inhibitory activity, as well as antifungal and antimicrobial effects against *Candida* fungi and multiple intraoral pathogenic microorganisms. Therefore, it may be a promising treatment option for oral infections.

## Introduction

Oral infections such as oral candidiasis and periodontal disease are a series of polymicrobial conditions that affect the oral mucosa and tooth root [1]. Deterioration of oral hygiene leads to intraoral colonization by fungi or bacteria, and has been reported to contribute to systemic diseases such as gingivitis, aspiration pneumonia, deep mycosis, or increased severity and mortality in corona virus disease-19 (COVID-19) [2–4]. It is suggested that aspiration pneumonia, which is often seen in the elderly, is caused by oral bacteria growing in the oral cavity. In addition to periodontal pathogens, opportunistic infectious pathogenic microorganisms such as *Candida*, *Pseudomonas*, or *Staphylococcus* are known to easily form multi-species biofilm coaggregations intraorally, causing various systemic conditions such as those associated with diabetes mellitus, cardiovascular diseases, pulmonary diseases, and preterm birth [3, 5, 6]. The elderly population has an increased incidence of aspiration pneumonia and fever due to aspiration of oral bacteria along with decreased immune function and swallowing reflex. In particular, elderly people who require nursing care are likely to have deteriorated oral hygiene conditions due to a decrease in independence and salivary secretion, thus having an increased risk of developing pneumonia. Therefore, maintaining oral hygiene through daily oral care is important in preventing aspiration pneumonia and oral infections. Oral care is attracting high interest in Japan, a country that has entered a super-aging society. *Candida* species are facultative anaerobes and can grow in an environment where oxygen is sufficiently supplied, such as the oral surface and tooth surface, and in an environment where oxygen concentration is low, such as between teeth and periodontal pockets. *Candida albicans* (*C. albicans*), *C. glabrata*, *C. krusei*, *C. parapsilosis*, and *C. tropicalis* have been isolated from the oral cavity. *C. albicans* is the most isolated species, but in recent years, there have been reports of infection by non-albicans *Candida* species [7, 8]. Proliferation of these fungi in the oral cavity can lead to superficial gingival inflammation, periodontitis, and deep-seated invasive gingival inflammation. [9]. Oral infection caused by *Candida* species is more likely to develop from immunodeficiency or decreased immunity owing to diabetes or human immunodeficiency virus infection [10–12]. In addition, unsanitary oral conditions have been reported to contribute to the development of Alzheimer's disease and/or increased severity and mortality in COVID-19 [13–18]. Hence, periodic oral examinations and maintenance of oral hygiene are important.

To prevent the exacerbation of poor oral hygiene, it is important to keep the oral cavity clean by brushing. In treating oral infections, cleansing the oral cavity through oral care using antiseptics, as well as treatment with antifungal drugs such as amphotericin B and miconazole is very important. However, *Candida* species are known to easily form biofilms on the surfaces of teeth and dentures, thereby reducing the effectiveness of brushing and antifungal agents [19–21]. The constituents of biofilms are polysaccharides and dead and live fungi [22]. Given

that biofilms have high adhesion to the tooth surface, it is difficult to remove them all by the self-cleaning action of saliva or brushing [23]. In addition, given that biofilms form a membranous structure, drugs found in mouthwashes do not penetrate the biofilm, reducing the effectiveness of the formulation [24, 25]. Therefore, suppressing the formation of biofilms on the surfaces of teeth and dentures is very important for the prevention and treatment of oral infectious diseases. Although antimicrobial agents such as chlorhexidine gluconate and/or cetylpyridinium chloride are used as additives in commercially available oral hygiene care products such as mouthwash and toothpaste, some of them are associated with side effects and cytotoxic [26, 27]. Plant derived-products have been used as oral care reagents that have fewer adverse effects [28]. Essential oil from *Matricaria chamomilla* has antimicrobial effects and mouthwash containing this oil suppressed oropharynx colonization by *Staphylococcus aureus* (*S. aureus*) and *Streptococcus pneumoniae* in patients in the intensive care unit [29], and cinnamon and sweet basil essential oils enhanced inhibition of biofilm formation by *Streptococcus mutans* and *Lactobacillus casei* [30].

Beta-thujaplicin (hinokitiol) is a bioactive compound of an aromatic seven-member tropolone purified from *Cupressaceae* plants such as *Chamaecyparis obtusa* and *Thujopsis dolabrata* [31, 32]. Hinokitiol has antimicrobial effects and has been used in many daily necessities, foods, and cosmetics owing to its low toxicity to humans [33–37]. However, the biological activity of this compound is not yet fully understood and there are few reports on its antimicrobial properties against opportunistic bacteria such as *S. aureus* or *Pseudomonas aeruginosa* (*P. aeruginosa*), which are rarely detected in oral cavities of healthy subjects but frequently detected in those of the elderly. In this study, we aimed to investigate whether oral care gels containing hinokitiol and 4-isopropyl-3-methylphenol (IPMP) (Hinora®: HO) or hinokitiol (Refrecare®: RC) have biofilm formation inhibitory and antimicrobial activities against fungi and bacteria that cause oral disease.

## Material and methods

### Strains and culture conditions

*C. albicans* (ATCC10231), *C. glabrata* (NBRC0622), *C. krusei* (NBRC1395), *C. parapsilosis* (NBRC1396), *C. tropicalis* (NBRC1400), *Aggregatibacter actinomycetemcomitans* (*A. actinomycetemcomitans*) (ATCC29522), *P. aeruginosa* (ATCC9027), and *S. aureus* (ATCC6538) were purchased from the American Type Culture Collection (ATCC, Virginia, USA) or Biological Resource Center, National Institute of Technology and Evaluation (NBRC, Chiba, Japan). All fungi and bacteria, except for *A. actinomycetemcomitans*, were cultured under aerobic conditions, and *A. actinomycetemcomitans* was cultured under anaerobic conditions (5% $CO_2$, 0% $O_2$), using Anaero Pouch Kenki (Mitsubishi Gas Chemicals Co., Ltd., Tokyo, Japan).

In the culture using soybean-casein digest (SCD) agar medium (Nissui pharmaceutical Co., Ltd., Tokyo, Japan), each fungus and bacteria were cultured at 37˚C for 24–48 h. For the culture using SCD broth (Merck KGaA, Darmstadt, Germany) with and without HO (EN Otsuka Pharmaceutical Co., Ltd., Iwate, Japan) and/or RC (EN Otsuka Pharmaceutical Co., Ltd., Iwate, Japan) (HO concentration: 0.05 g/mL, 0.1 g/mL; RC concentration: 0.05 g/mL, 0.1 g/mL), the obtained fungal and bacterial suspensions were dispensed onto 96-well polystyrene plates at a density of 200 μL/well and cultured at 37˚C for 24–48 h. For all experiments, each fungus was adjusted to an optical density (OD) of 0.1–0.4 at 600 nm (equivalent to $1 \times 10^6$ colony-forming units (CFU) /mL) and inoculated at $1 \times 10^6$ CFU/mL, and each bacterium was adjusted to an OD of 0.1–0.3 at 600 nm (equivalent to $1 \times 10^8$ CFU/mL) and inoculated at $1 \times 10^8$ CFU/mL.

## Antimicrobial disc diffusion assay

Each fungus and bacteria were inoculated into saline solutions to achieve a concentration of $1 \times 10^6$ or $1 \times 10^8$ CFU/mL and spread evenly on SCD agar medium. A sterilized paper disc with a diameter of 8 mm (ADVANTEC Toyo, Co., Ltd., Tokyo, Japan) coated with oral care gel-containing medium was placed on the SCD agar medium, and the fungus or bacterium was cultured at 37°C for 24 h.

Paper discs were coated with 80 μL of SCD broth and SCD broth containing HO (0.1, 0.5, and 5.0 g/mL and undiluted form) or RC (0.1, 0.5, and 5.0 g/mL and undiluted form). A paper disc coated with oral care gel-containing medium was used after 2–3 h of air drying on a clean workstation.

## Microorganism turbidity and crystal violet assays

As described above, each fungus and bacteria were cultured at 37°C for 24–48 h in SCD broth and SCD broth containing HO (0.05 g/mL, 0.1 g/mL) or RC (0.05 g/mL, 0.1 g/mL). Fungal or bacterial growth was determined to be proportional to the measured OD at 600 nm by spectrophotometry. First, the absorbance of cultures was measured at 600 nm using a SpectraMax190 microplate reader (Molecular Devices Co., Ltd., Tokyo, Japan) to monitor the turbidity (i.e., possibility of proliferation of each fungus and bacterium).

Next, the medium was removed, and each well was washed twice with saline. A crystal violet aqueous solution (0.1% w/v) was added to each well (200 μL/well), and the mixture was allowed to stand at room temperature (~ 25°C) for 30 min for staining. Crystal violet is a basic purple dye having a triphenylmethane skeleton, and has been widely used for the purpose of staining the cell wall or biofilm of fungi and bacteria. It is suitable for biofilm staining at OD 590 nm [38]. Generally, biofilms are formed 24–48 h after culturing the fungus or bacterium. Herein, after staining, the crystal violet aqueous solution was removed, and each well was washed twice with saline. Ethanol was added to the well (200 μL/well), and the mixture was allowed to stand at approximately 25°C for 15 min. Then, the crystal violet stain solution was extracted from each fungus or bacterium with ethanol, and the OD was measured at 590 nm using a microplate reader to detect the biofilm in the 96-well plate.

## Germ tube test

The germ tube test was performed using a modification of Mackenzie's method [39]. *C. albicans* was inoculated into SCD broth or SCD broth with HO (0.05 g/mL) to a concentration of $1 \times 10^6$ CFU/mL. The fungal suspension was applied in a 6-cm dish (3 mL/dish) and cultured at 37°C for 3 h. Images for morphological observation of the germ tubes were acquired with an Olympus CKX41 camera (Olympus Co., Ltd., Tokyo, Japan). The number of germ tube-forming cells per 500 cells were divided into three sections and counted (1,500 cells in total), and compared between the control group and the HO-added group.

## Statistical analysis

The statistical significance of the differences in the ratio of biofilm formation between non-treated, and HO- and RC-treated fungal or bacterial pathogens was examined using Student *t* test or Dunnett's test. The statistical significance level was considered to be statistically significant when $P < 0.05$.

## Results

The inhibitory zone formed by the undiluted solution of HO and RC for each *Candida* fungus was larger than that of the control group (Fig 1). The inhibition zone was larger for HO than

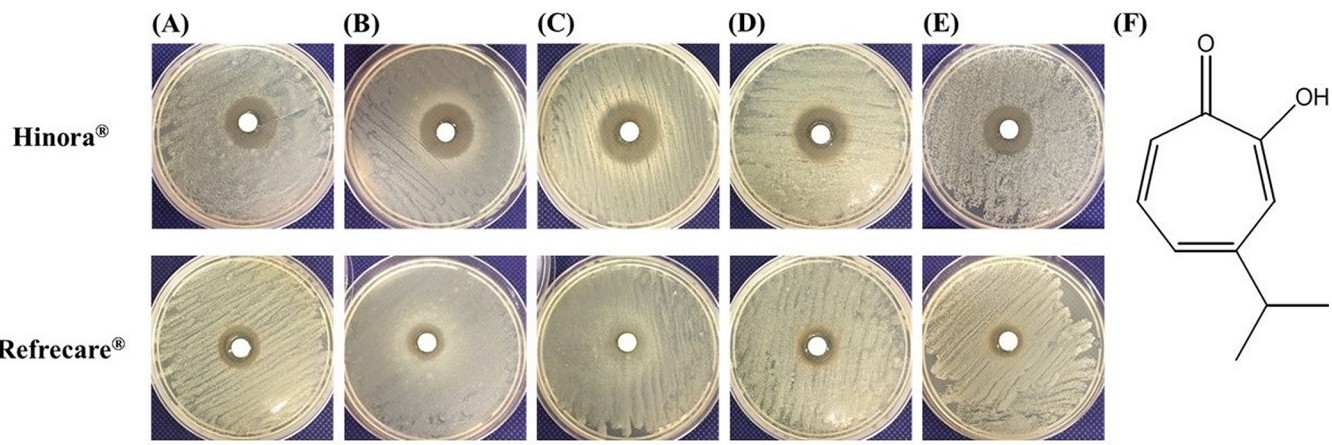

**Fig 1.**

for RC, with diameters of 24 mm (HO), 19 mm (RC) for *C. albicans* (Fig 1A), 25 mm (HO), 15 mm (RC) for *C. glabrata* (Fig 1B), 25 mm (HO), 10 mm (RC) for *C. krusei* (Fig 1C), 24 mm (HO), 17 mm (RC) for *C. parapsilosis* (Fig 1D), and 23 mm (HO), 14 mm (RC) for *C. tropicalis* (Fig 1E). In addition, other oral infection-causing microorganisms used in the experiments formed inhibitory zones, with diameters of 25 mm (HO), 13 mm (RC) for *A. actinomycetemcomitan*, 31 mm (HO), 20.5 mm (RC) for *S. aureus*, and 10 mm (HO), 15 mm (RC) for *P. aeruginosa* (Table 1). The maximum zone of inhibition was 31 mm of *S. aureus* in the HO-added group. In the HO-added group, inhibition circles were observed for all fungi and bacteria except *P. aeruginosa* at a concentration of 0.5 g/mL or higher. In the RC-added group, inhibition circles were observed for all fungi and bacteria except *C. krusei* at a concentration of 5.0 g/mL or higher (Table 1).

Next, *Candida* species and bacteria were cultured in SCD broth containing HO or RC (final 0.05, 0.1 g/mL) for 24–48 h, and each fungal or bacterial growth was measured using spectrophotometry by absorbance at 600 nm OD. The OD at 600 nm was significantly lower when cultured in SCD broth containing HO or RC than that observed for the control (Fig 2). These

**Table 1. Inhibition zone diameter measurement (mm) in *Candida* spp., *Aggregatibacter actinomycetemcomitans*, *Staphylococcus aureus*, and *Pseudomonas aeruginosa* by Hinora® (HO) and Refrecare® (RC).**

| Oral care gel containing hinokitiol (g/mL) | | Inhibition zone diameter (mm), including a sterilized paper disc with a diameter of 8 mm (n = 1) | | | | | | | |
|---|---|---|---|---|---|---|---|---|---|
| | | *C. albicans* | *C. glabrata* | *C. krusei* | *C. parapsilosis* | *C. tropicalis* | *A. actinomy* | *S. aureus* | *P. aeruginosa* |
| Hinora® | 0 | ND | ND | ND | ND | ND | ND | ND | ND |
| | 0.1 | ND | ND | ND | ND | ND | ND | 12 | ND |
| | 0.5 | 13 | 14 | 13 | 12 | 11 | 15 | 22.5 | ND |
| | 5.0 | 18 | 22 | 15 | 22 | 16 | 19 | 26.5 | ND |
| | undiluted form | 24 | 25 | 25 | 24 | 23 | 25 | 31 | 10 |
| Refrecare® | 0 | ND | ND | ND | ND | ND | ND | ND | ND |
| | 0.1 | ND | ND | ND | ND | ND | ND | 9.5 | ND |
| | 0.5 | ND | ND | ND | ND | ND | ND | 16 | ND |
| | 5.0 | 10 | 10 | ND | 10 | 12 | 9.5 | 20 | 11 |
| | undiluted form | 19 | 15 | 10 | 17 | 14 | 13 | 20.5 | 15 |

ND: not detectable

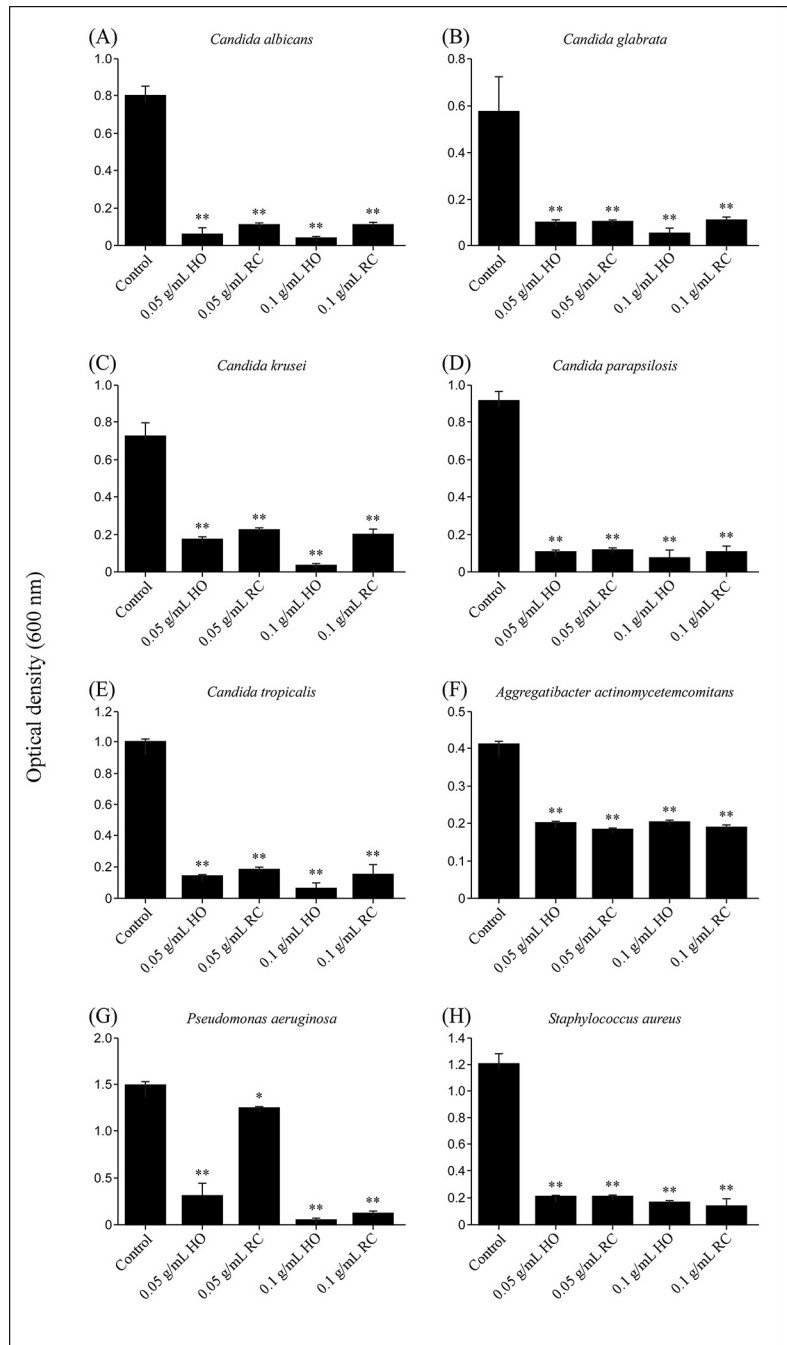

**Fig 2.**

growth suppression effects were observed when the concentration of HO or RC was 0.05 g/mL or higher.

Subsequently, we investigated whether these oral care gels had biofilm formation inhibitory activity against each fungus and bacteria using crystal violet assays. As described above, each fungus and bacteria were cultured, and each fungal and bacterial biofilm formation was measured using spectrophotometry by absorbance at 590 nm OD. The OD at 590 nm was significantly lower when cultured in SCD broth containing HO or RC than that observed for the

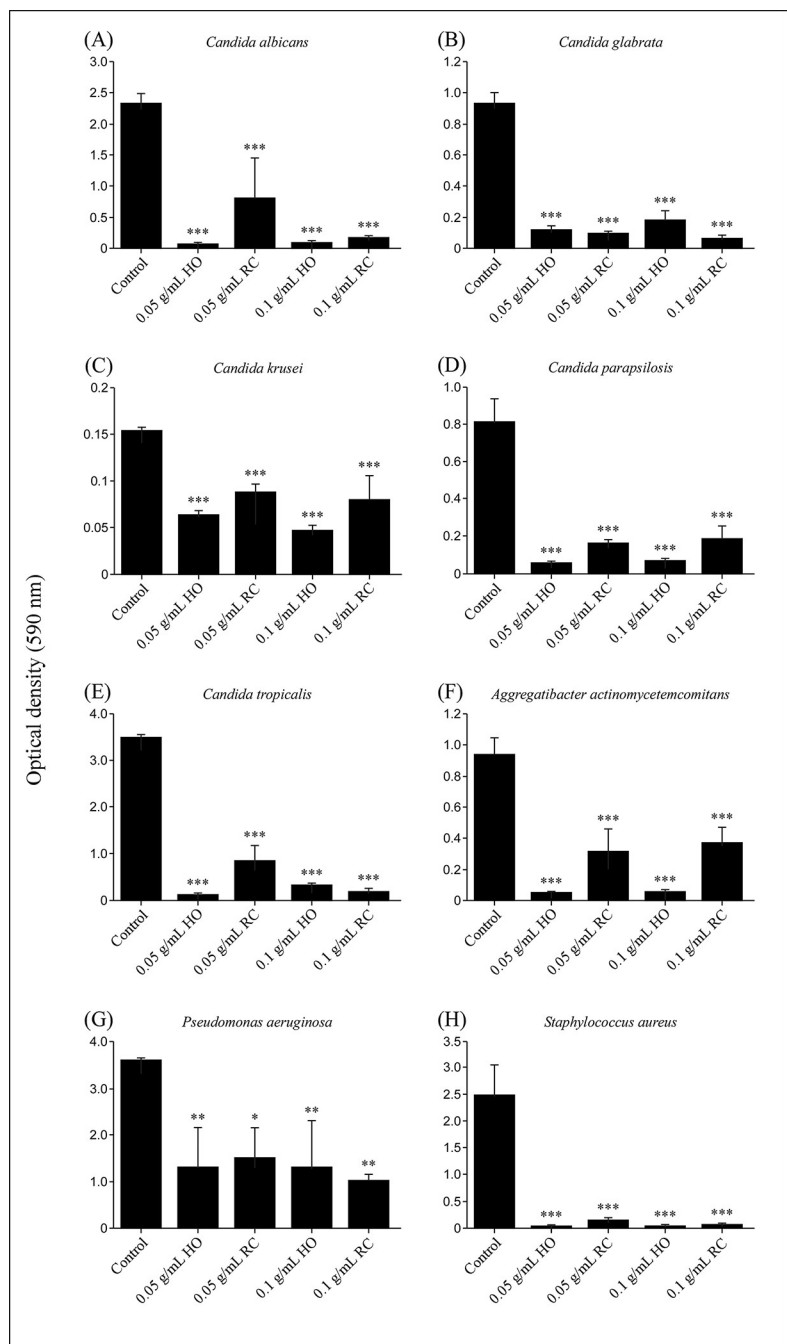

**Fig 3.**

control (Fig 3). The addition of HO inhibited biofilm formation of all the oral infection-causing microorganisms used. The inhibitory effect of HO was observed at 0.05 g/mL or higher. RC also showed significant biofilm formation inhibitory effect against *Candida* species, periodontal disease bacteria, and opportunistic infectious bacteria at above 0.05 g/mL, but the effect tended to be weaker than that of HO. In contrast, HO and RC were equally effective in inhibiting biofilm formation against *P. aeruginosa*.

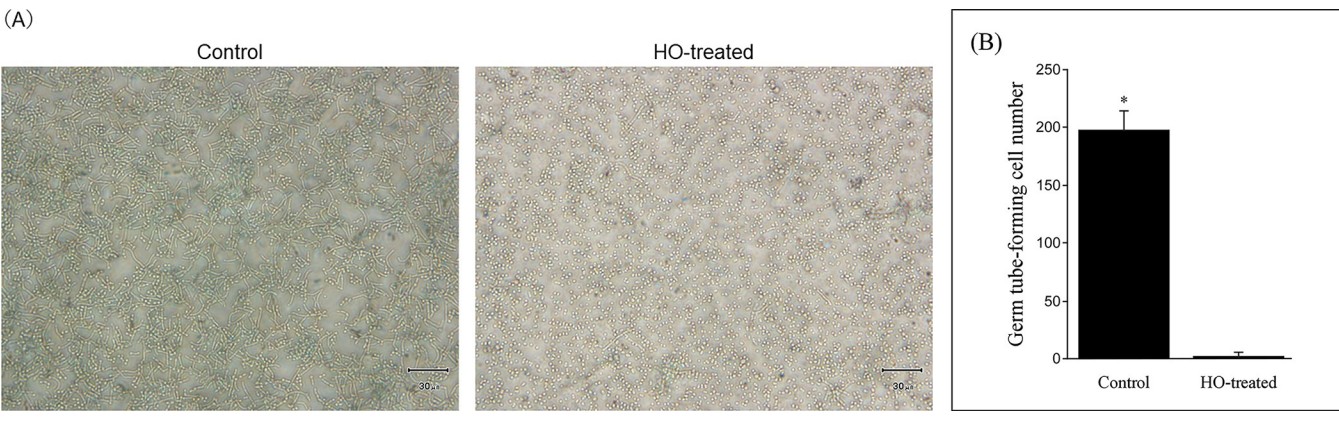

**Fig 4.**

Finally, we conducted a germ tube test of *C. albicans* using HO-containing SCD broth to investigate the mechanism of action of HO, which had an excellent ability to inhibit biofilm formation. The addition of HO significantly suppressed the formation of germ tubes in *C. albicans* (Fig 4A). Germ tube formation was inhibited even at a concentration as low as 0.05 g/mL HO, and the total number of germ tube-forming cells in 1,500 cells was significantly reduced in the HO-added group (1.7 ± 1.7/500 cells) compared to the control group (198 ± 17.1/500 cells) ($P < 0.01$) (Fig 4B).

## Discussion

Oral care is important not only for disease prevention or functional maintenance of teeth and surrounding tissues, but also for prevention of systemic diseases and maintenance of quality of life. In Japan, which is currently a super-aging society, in addition to oral care aimed at preventing caries and periodontal disease, the need for oral care for the elderly and those requiring nursing care is increasing [40]. Although there have been previous reports on the antimicrobial properties of hinokitiol, there are few reports that have extensively verified its antimicrobial effects against various microbes closely related to oral infections. Investigation of the antimicrobial properties of oral care products containing hinokitiol revealed the presence of antimicrobial activity against *C. albicans* (ATCC10231), *C. glabrata* (NBRC0622), *C. krusei* (NBRC1395), *C. parapsilosis* (NBRC1396), and *C. tropicalis* (NBRC1400), which are the main cause of oral candidiasis, at low concentration (0.05 g/mL). Furthermore, similar effects were obtained against *P. aeruginosa*, *S. aureus*, and periodontal disease bacteria (Table 1). Fig 1 shows the growth inhibition zones of *Candida* species. For all fungi and bacteria that cause oral infectious diseases, growth-inhibiting circles were formed around the disc containing HO or RC (Table 1). In the HO-added group, inhibition circles were observed in all bacteria except *P. aeruginosa* when added at a concentration of 0.5 g/mL or more. In the RC-added group, inhibition circles were observed in all bacteria except *C. krusei* when added at a concentration of 5.0 g/mL or more (Table 1). In addition, various fungi and bacteria described above were cultured in SCD broth with and/or without HO or RC for 24 h, and the OD of the culture was measured at 600 nm on a microplate reader to detect the turbidity (indicative of the level of proliferation of each fungus and bacteria). Growth inhibition was observed when HO or RC was added at a concentration of above 0.05 g/mL (Fig 2). The OD values at 600 nm of all the cultures were significantly reduced when cultured in media containing HO or RC. These results indicate that HO and RC have growth inhibitory effects on *Candida* species, as well as

opportunistic and periodontal disease bacteria, and support the efficacy of hinokitiol in the dental field. The results of the disc diffusion assay and culture in SCD liquid medium showed that HO had higher antimicrobial activity than RC. Hinokitiol content is twice as much in HO when compared to that in RC (HO: 0.1%; RC: 0.05%). Moreover, 0.02% 4-isopropyl-3-methyl-phenol (IPMP) is present in HO but not in RC. IPMP is a typical preservative and bactericidal constituent in cosmetics and is used in commercial oral care products [41]. Phenolic compounds such as IPMP, phenoxyethanol, and ortho-phenylphenol are often thought to exhibit antimicrobial activity by damaging cell membranes and changing the membrane structure and function [42]. These facts are consistent with the observation that HO had higher antimicrobial activity than RC.

Biofilms provide optimal protection for embedded cells from antibiotics, antifungals, and the immune system, which is the main reason why biofilm-related infections are difficult to treat. In addition, microorganisms on the surface of biofilms can spread to the infectious region as they move and multiply in search of other habitats as needed [22, 43, 44]. We investigated whether oral care gels, HO and RC, could inhibit biofilm formation caused by oral infection-causing microorganisms. The addition of HO or RC at concentrations above 0.05 g/mL inhibited biofilm formation of *Candida* species and all oral infection-causing organisms studied (Fig 3). Biofilm formation inhibitory effects of intraoral microorganisms were stronger for HO than RC. Hence, daily use of oral care products containing hinokitiol and IPMP is thought to contribute to reducing the risk of intraoral infections.

In elderly people who require nursing care, more opportunistic bacteria such as *Candida* species, *P. aeruginosa*, *S. aureus*, and periodontal disease bacteria are detected in the oral cavity due to the wearing of dentures and the deterioration of biological defenses [3, 6]. In addition, a decrease in self-cleaning action due to a decrease in salivary secretion also leads to deterioration of oral hygiene. In particular, the proliferation of *Candida* species is closely related to the risk of death from respiratory diseases and the development of aspiration pneumonia.

Germ tube formation is one of the processes involved in adhesion, which is the first step in the biofilm formation of *Candida* species [23, 45]. Therefore, inhibition of germ tube expression contributes to the control of *Candida* species infection. The formation of germ tubes is known to be promoted by stimulation with proteins in serum at 37°C and is easily formed even in SCD broth [39, 46]. In this study, to investigate the mechanism of action of HO, *C. albicans* was cultured in HO-containing SCD broth for 3 h, and its effect on germ tube formation was investigated. The results showed that culturing *C. albicans* in SCD broth with HO (0.05 g/mL) significantly suppressed germ tube formation (Fig 4). Therefore, oral care products containing hinokitiol can be expected to suppress biofilm formation of *Candida* species by inhibiting the germ tube formation. Hinokitiol-containing oral care gel exerts a strong biofilm formation inhibitory effect and antifungal activity against *Candida* species, albeit at a low concentration (0.05 g/mL) (Figs 3 and 4). Furthermore, this study revealed that hinokitiol also exhibits antimicrobial activity against *P. aeruginosa*, *S. aureus*, and periodontal disease bacteria, findings which have not been reported to date. These data suggest that the use of oral care products containing hinokitiol in oral care for the elderly requiring nursing care may prevent oral infections and contribute to the improvement of their quality of life.

It is speculated that hinokitiol inhibits biofilm formation by suppressing the expression of genes and proteins related to hyphal formation and adhesion to biogenic organisms. Kim et al. reported that hinokitiol suppresses the expression of genes involved in the adhesion process and hyphal formation and/or maintenance of *C. albicans*, resulting in the inhibition of biofilm formation [47]. However, whether a similar biofilm inhibition mechanism applies in non-albicans *Candida* species such as *S. aureus* and *P. aeruginosa* has not been demonstrated. In the future, in addition to gene expression analysis related to biofilms, morphological imaging

measurements using field emission scanning electron microscopy or high-resolution transmission electron microscopy will be necessary for further elucidation of the anti-biofilm mechanism of hinokitiol, and investigation is required to evaluate the antimicrobial activity of hinokitiol *in vivo*.

## Conclusion

In conclusion, oral care gel products containing hinokitiol were found to have biofilm formation inhibitory and antimicrobial activities against various intraoral pathogenic microorganisms. In particular, the oral care gel containing hinokitiol and IPMP strongly inhibited biofilm formation and had antimicrobial effects against multiple oral infection-causing microorganisms. These results suggest that oral care gel products containing hinokitiol are very effective in the prevention and treatment of oral infections such as oral candidiasis or periodontal disease. In addition, oral care gel products containing hinokitiol and IPMP may be particularly useful for assisting with the prevention and treatment of multiple oral infectious microorganisms.

## Acknowledgments

We are also grateful to Kenichi Ogasawara and Ryosuke Kawawaki of EN Otsuka Pharmaceutical Co., Ltd. for their advice in promoting this research.

## Author Contributions

**Data curation:** Hiroshi Ohara, Keita Odanaka.

**Formal analysis:** Hiroshi Ohara.

**Funding acquisition:** Masataka Hayasaka.

**Investigation:** Hiroshi Ohara, Miku Shiine.

**Methodology:** Hiroshi Ohara, Keita Odanaka.

**Project administration:** Hiroshi Ohara.

**Supervision:** Hiroshi Ohara, Masataka Hayasaka.

**Validation:** Hiroshi Ohara, Miku Shiine.

**Writing – original draft:** Hiroshi Ohara.

**Writing – review & editing:** Hiroshi Ohara, Keita Odanaka.

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
