## [Decision Letter · Decision Letter 0]

31 Mar 2023

PONE-D-23-06120Antibacterial Effect of Oral Care Gel-Containing Hinokitiol and 4-isopropyl-3-methylphenol Against Intraoral Pathogenic MicroorganismsPLOS ONE

Dear Dr. Ohara,

Thank you for submitting your manuscript to PLOS ONE. After careful consideration, we feel that it has merit but does not fully meet PLOS ONE’s publication criteria as it currently stands. Therefore, we invite you to submit a revised version of the manuscript that addresses the points raised during the review process.

We look forward to receiving your revised manuscript.

Kind regards,

José António Baptista Machado Soares, PhD

Academic Editor

PLOS ONE

Journal Requirements:

Additional Editor Comments (if provided):

Dear authors,

The present work is very interesting. However, both reviewers appointed several concerns about the original version of the manuscript. Briefly, both reviewers required English editing and double-checking of results (figures and table) and rewriting of most sections. Please carefully answer all concerns raised by the reviewers.

Reviewers' comments:

Reviewer's Responses to Questions

**Comments to the Author**

1. Is the manuscript technically sound, and do the data support the conclusions?

Reviewer #1: Partly

Reviewer #2: Yes

2. Has the statistical analysis been performed appropriately and rigorously? 

Reviewer #1: Yes

Reviewer #2: Yes

3. Have the authors made all data underlying the findings in their manuscript fully available?

Reviewer #1: Yes

Reviewer #2: Yes

4. Is the manuscript presented in an intelligible fashion and written in standard English?

Reviewer #1: No

Reviewer #2: Yes

5. Review Comments to the Author

Reviewer #1: After the reviewing the submitted manuscript, it must be carefully revised as the following:

1-The title must be more attractive and changed to antimicrobial instead of antibacterial.

2-Introduction must be reconstruct to be more informative and contains tge problem and how the other publicatuons try to solve this problem and how your paper try to solve that.

3-Deep discussion and related and recent studies must br included.

4-The novelty and the future perspective in addition to limitation must be included.

5-Antibiofilm assay must be included in addition to growth kinetics.

6-Reaction mechanism determinant must be performed by FESEM, HRTEM, and membrane leakage assay.

7-English Language editing must be performed by English Native Speaker.

Reviewer #2: The manuscript entitled “Antibacterial Effect of Oral Care Gel-Containing Hinokitiol and 4-isopropyl-3-methylphenol Against Intraoral Pathogenic Microorganisms” is about to provide the evidence for new anti-intraoral pathogenic microorganisms agents.

After first time use, authors can use abbreviation for microorganism’s scientific name. I found Candida albicans several times.

At “Antimicrobial disc diffusion assay”, it is not clear why authors use SCD medium? Why there is no reference?

At the crystal violet assays, it is not clear how to induce the biofilm formation, so authors need to explain more clearly.

At the “Germ tube test”, it is not clear the media is appropriate to induce the germ tube. Please check again.

At line 166, OD may not correct in there. So, check and change.

Authors frequently use bacteria even though the assay include fungi. So, authors need to change that.

At results, authors need to explain the results with the number to make easy to understand.

At discussion, Turbidity is not directly indicative for the level of proliferation. So, authors need to change that.

At discussion, discussion is written usually for explain the results, so if authors add more possible mechanism of HO, that should be a better discussion.

If the HO is a gel-type, how authors weighing the HO? because weight of HO just includes hinokitiol and IPMP, or include hinokitiol and IPMP with some others?

Why authors mixed the hinokitiol and IPMP and what is the ratio?

At figure 1, the figure captured were compared for each bacterium correctly? I think authors need to double check that.

At figure 4, authors need to provide that results statistically and those are hyphae or pseudohyphae? I suggested that authors need to explain about that.

Finally, I could not find table.

6. PLOS authors have the option to publish the peer review history of their article (what does this mean?). If published, this will include your full peer review and any attached files.

Reviewer #1: **Yes: **Prof. Gharieb S. El-Sayyad

Reviewer #2: No

---

## [Author Response · Author response to Decision Letter 0]

3 Jul 2023

Response to reviewers and editor comments

We are grateful to the reviewers and editor for their helpful comments that have contributed to the improvement of our research paper. The manuscript has benefited from these insightful suggestions. We have provided the responses to the comments below:

Reviewer #1: After the reviewing the submitted manuscript, it must be carefully revised as the following:

1-The title must be more attractive and changed to antimicrobial instead of antibacterial.

→Thank you for pointing it out. The title has been changed according to your suggestion.

2-Introduction must be reconstructed to be more informative and contains the problem and how the other publications try to solve this problem and how your paper try to solve that.

→Thank you for your advice. In lines 57–68 and 93–101 of the introduction, we have included the current problem(s) and the solutions from previous studies and our attempts.

3-Deep discussion and related and recent studies must include.

→We have cited a recent research paper in lines 93–101 of the introduction. Moreover, we have included recent studies in the discussion in lines 218–230, 249–251, and 253–270 of the discussion.

4-The novelty and the future perspective in addition to limitation must be included.

→Thank you for your comment. We have revised the discussion and added the novelty and limitations of the research in lines 271–287 and 292–298, respectively.

5-Antibiofilm assay must be included in addition to growth kinetics.

→Thank you for your suggestion. We have added a description of the anti-biofilm assay in the Material and methods section, lines 154–157. In addition, we have added considerations on anti-biofilm assays in lines 253–263 of the discussion.

6-Reaction mechanism determinant must be performed by FESEM, HRTEM, and membrane leakage assay.

→Thank you for your advice. You are absolutely right; we also think that it is necessary to investigate reaction mechanism using FE-SEM and HR-TEM. However, currently there is a limit to what we can do with our current research equipment and funding; therefore, we were unable to perform these experiments. This is a limitation of our research and has been included as a subject for future study in lines 294–298.

7-English Language editing must be performed by English Native Speaker.

→ English language editing has been performed by an English Native Speaker.

Reviewer #2: The manuscript entitled “Antibacterial Effect of Oral Care Gel-Containing Hinokitiol and 4-isopropyl-3-methylphenol Against Intraoral Pathogenic Microorganisms” is about to provide the evidence for new anti-intraoral pathogenic microorganism agents.

After first time use, authors can use abbreviation for microorganism’s scientific name. I found Candida albicans several times.

→We have corrected the relevant parts in the entire manuscript.

At “Antimicrobial disc diffusion assay”, it is not clear why authors use SCD medium? 

Why there is no reference?

→Thank you for your comment. SCD medium is commonly used as a medium for Candida species; therefore, to unify the experimental conditions in this study, SCD medium was used. Sufficient growth was observed in SCD medium for all fungi and bacterium used herein.

At the crystal violet assays, it is not clear how to induce the biofilm formation, so authors need to explain more clearly.

→We have included information on biofilm formation in the Material and methods in lines 154–158.

At the “Germ tube test”, it is not clear the media is appropriate to induce the germ tube. Please check again.

→Thank you for your advice. We have checked our data again, and since significant formation of germ tubes was observed in C. albicans cultured in SCD broth, we believe that the medium used in this study is suitable for formation of germ tubes.

At line 166, OD may not correct in there. So, check and change.

→We have double-checked the OD values and added/corrected these in lines 202–204.

Authors frequently use bacteria even though the assay include fungi. So, authors need to change that.

→Thank you for your comment. We have carefully corrected this in the entire manuscript by distinguishing between fungi and bacterium.

At results, authors need to explain the results with the number to make easy to understand.

→We added a description of the results numerically in lines 181–187 and 214–216.

At discussion, Turbidity is not directly indicative for the level of proliferation. So, authors need to change that.

→Thank you for your comment. We have presented turbidity to simply refer to "possibility of proliferation" in lines 148–151.

At discussion, discussion is written usually for explain the results, so if authors add more possible mechanism of HO, that should be a better discussion.

→Thank you for your advice. We mentioned the possible mechanism of HO in the Discussion in lines 288–294.

If the HO is a gel-type, how authors weighing the HO? because weight of HO just includes hinokitiol and IPMP, or include hinokitiol and IPMP with some others?

→Thank you for your question. HO in the tube was placed directly into a sterile 15 mL or 50 mL centrifuge tube and weighed using an electronic balance. In addition to hinokitiol and IPMP, HO contains an anti-inflammatory ingredient (glycyrrhizinate dipotassium), a solvent (purified water), a humectant system (concentrated glycerin, propylene glycol, hydrolyzed collagen, sodium hyaluronate), a solubilizer (polyoxyethylene hydrogenated castor oil), a viscosity adjuster (carboxyvinyl polymer, sodium polyacrylate), a pH adjuster (potassium hydroxide), a preservative (sodium benzoate), a flavoring agent (flavor), and a stabilizer (ethylenediaminetetraacetic acid disodium).

Why authors mixed the hinokitiol and IPMP and what is the ratio?

→Hinola® (HO) is an oral care gel developed by Otsuka Pharmaceutical Co., Ltd. Hinokitiol, which is extracted from Cupressaceae plants. It has long been used in daily life for its insect repellent, deodorant, and antibacterial effects. IPMP has broad-spectrum bactericidal properties and is effective against bacteria or fungi. For example, it is often used in acne care products to suppress acne-causing bacteria, and in antiperspirants to suppress the growth of odor-causing bacteria. These two ingredients are still used in various common daily products today, and have been applied in the dental field, leading to the development and commercialization of Hinola® (0.1% hinokitiol and 0.02% IPMP).

At figure 1, the figure captured were compared for each bacterium correctly? I think authors need to double check that.

→Thank you for pointing this out. The acquisition and measurement of the images in figure 1 were all performed under the same conditions. We also reconfirmed that the images captured were correctly compared for each bacterium.

At figure 4, authors need to provide that results statistically and those are hyphae or pseudohyphae? I suggested that authors need to explain about that.

→Thank you for your suggestion. We compared the number of germ tube-producing cells in 500 cells between the control group and the HO-added group, and performed this in three sections. We next statistically analyzed and graphed the results (Fig. 4B).

Candida albicans cultured in SCD broth formed biofilms (Fig. 3). Since hyphae are necessary for biofilm formation, we believe that the images are hyphae rather than pseudohyphae (Fig. 4A). We mentioned this in lines 271–272 of the Discussion.

Finally, I could not find table.

→Please accept our sincerest apologies. We have uploaded the missing table.

---

## [Decision Letter · Decision Letter 1]

31 Jul 2023

PONE-D-23-06120R1Antimicrobial Effect of Oral Care Gel-Containing Hinokitiol and 4-isopropyl-3-methylphenol Against Intraoral Pathogenic MicroorganismsPLOS ONE Thank you for submitting your manuscript to PLOS ONE. After careful consideration, we feel that it has merit but does not fully meet PLOS ONE’s publication criteria as it currently stands. Therefore, we invite you to submit a revised version of the manuscript that addresses the points raised during the review process.

We look forward to receiving your revised manuscript.

Dear authors,

I am pleased to say that the reviewers enjoyed the manuscript very much and we are excited about the possibility to publish your work. One reviewer already endorsed the revised manuscript for publication, while another reviewer asked for clarification about Table 1 in lines 192 and 236 and to increase the resolution of all figures. So, I kindly invite the authors to realize these minor rectifications to finally endorse your study.

Thank you and best regards,

António Machado

Reviewers' comments:

Reviewer's Responses to Questions

**Comments to the Author**

1. If the authors have adequately addressed your comments raised in a previous round of review and you feel that this manuscript is now acceptable for publication, you may indicate that here to bypass the “Comments to the Author” section, enter your conflict of interest statement in the “Confidential to Editor” section, and submit your "Accept" recommendation.

Reviewer #1: All comments have been addressed

Reviewer #2: (No Response)

2. Is the manuscript technically sound, and do the data support the conclusions?

Reviewer #1: No

Reviewer #2: Yes

3. Has the statistical analysis been performed appropriately and rigorously? 

Reviewer #1: Yes

Reviewer #2: Yes

4. Have the authors made all data underlying the findings in their manuscript fully available?

Reviewer #1: Yes

Reviewer #2: Yes

5. Is the manuscript presented in an intelligible fashion and written in standard English?

Reviewer #1: Yes

Reviewer #2: Yes

6. Review Comments to the Author

Reviewer #1: After the complete revision of the submitted manuscript and following the authors response, the manuscript can be accepted for publication.

Reviewer #2: There is a table 1 in line 192 and 236, but I could not find table1.

Authors need to increase the resolution of all figure.

7. PLOS authors have the option to publish the peer review history of their article (what does this mean?). If published, this will include your full peer review and any attached files.

Reviewer #1: **Yes: **Prof. Dr. Gharieb S. El-Sayyad

Reviewer #2: No

---

## [Author Response · Author response to Decision Letter 1]

8 Aug 2023

Response to Reviewers

We sincerely appreciate to the reviewers for their supports that have contributed to the improvement of our research paper. We have provided the responses to the comments below:

Reviewer #2: There is a table 1 in line 192 and 236, but I could not find table1.

Authors need to increase the resolution of all figure.

→Please accept our sincerest apologies. We have included table 1 as part of our main manuscript after "References". In addition, we have tried to improve the resolution of the all figures as much as possible. We kindly ask for your confirmation.

---

## [Decision Letter · Decision Letter 2]

22 Aug 2023

Antimicrobial Effect of Oral Care Gel-Containing Hinokitiol and 4-isopropyl-3-methylphenol Against Intraoral Pathogenic Microorganisms

PONE-D-23-06120R2

Dear Dr. Ohara,

We’re pleased to inform you that your manuscript has been judged scientifically suitable for publication and will be formally accepted for publication once it meets all outstanding technical requirements.

Kind regards,

José António Baptista Machado Soares, PhD

Academic Editor

PLOS ONE

Reviewers' comments:

Reviewer's Responses to Questions

**Comments to the Author**

1. If the authors have adequately addressed your comments raised in a previous round of review and you feel that this manuscript is now acceptable for publication, you may indicate that here to bypass the “Comments to the Author” section, enter your conflict of interest statement in the “Confidential to Editor” section, and submit your "Accept" recommendation.

Reviewer #2: All comments have been addressed

2. Is the manuscript technically sound, and do the data support the conclusions?

Reviewer #2: Yes

3. Has the statistical analysis been performed appropriately and rigorously? 

Reviewer #2: Yes

4. Have the authors made all data underlying the findings in their manuscript fully available?

Reviewer #2: Yes

5. Is the manuscript presented in an intelligible fashion and written in standard English?

Reviewer #2: Yes

6. Review Comments to the Author

Reviewer #2: (No Response)

7. PLOS authors have the option to publish the peer review history of their article (what does this mean?). If published, this will include your full peer review and any attached files.

Reviewer #2: No

---

## [Editor Report · Acceptance letter]

24 Aug 2023

PONE-D-23-06120R2 

Antimicrobial Effect of Oral Care Gel Containing Hinokitiol and 4-isopropyl-3-methylphenol Against Intraoral Pathogenic Microorganisms 

Dear Dr. Ohara:

I'm pleased to inform you that your manuscript has been deemed suitable for publication in PLOS ONE. Congratulations! Your manuscript is now with our production department. 

Kind regards, 

on behalf of

Dr. José António Baptista Machado Soares 

Academic Editor

PLOS ONE